# 14-3-3 proteins: Regulators of cardiac excitation–contraction coupling and stress responses

**Heather C. Spooner**,
**Rose E. Dixon**

Department of Physiology and Membrane Biology, University of California Davis, School of Medicine, Davis, CA, USA

## Abstract

14-3-3 proteins are highly conserved proteins that regulate numerous cellular processes mostly through phosphorylation-dependent protein–protein interactions. In the heart 14-3-3 proteins play critical roles in cardiac conduction pathways, excitation–contraction (EC) coupling, development and stress responses. This review summarizes the current understanding of cardiac 14-3-3 regulation and function, with particular emphasis on its role in ion channel regulation and $\beta$-adrenergic signalling. We discuss how 14-3-3 proteins act through three main mechanisms – masking, clamping, and scaffolding – to regulate target proteins, including Cx43, $Ca_V1.2$, $Na_V1.5$, and various potassium channels. The seven mammalian 14-3-3 isoforms display distinct but overlapping functions, with tissue-specific expression patterns and isoform-specific regulation through phosphorylation and dimerization. Recent work has revealed 14-3-3's importance in cardiac development and stress responses, where it generally serves a cardioprotective role. However in some pathological contexts such as ischaemia–reperfusion injury, 14-3-3 can be detrimental. We highlight emerging themes in cardiac 14-3-3 biology, including its role in prolonging $\beta$-adrenergic signalling. Understanding the complex regulation of cardiac 14-3-3 and its numerous targets presents both opportunities and challenges for therapeutic development.

## Graphical Abstract

**Corresponding author** Rose E. Dixon, Department of Physiology and Membrane Biology, 4453 Tupper Hall, One Shields Avenue, University of California Davis School of Medicine, Davis, CA 95616, USA. redickson@ucdavis.edu.

Author contributions

H.C.S. and R.E.D. conceptualized the work. H.C.S. prepared the original draft, which R.E.D. subsequently expanded. Both authors edited the manuscript to its current form. All authors have approved the final version of the manuscript and agreed to be accountable for all aspects of the work. All persons designated as authors qualify for authorship, and all those who qualify for authorship are listed.

Additional information

Competing interests

The authors declare that they have no competing interests.

14-3-3 protein interactions in cardiac regulation. Schematic representation of 14-3-3 binding partners in excitation–contraction coupling, transcriptional regulation/development and stress response pathways. Asterisks indicate targets where the exact 14-3-3 binding site is unknown.

## Keywords

14-3-3; cardiovascular physiology; cardiac stress; EC coupling; ion channel regulation; ion channel clustering

## Introduction

14-3-3 proteins are small (27–32 kDa), ubiquitously expressed proteins that are highly conserved across eukaryotic organisms. They function as molecular adaptors that interact with hundreds of mostly phosphorylated binding partners to regulate diverse cellular processes, including cell cycle progression, protein trafficking, ion channel function and cellular stress responses. In humans seven distinct isoforms ($\beta$, $\epsilon$, $\gamma$, $\eta$, $\tau$, $\zeta$ and $\sigma$) are encoded by genes located on different chromosomes, all of which are expressed in the healthy heart with the exception of 14-3-3 $\sigma$. The unusual name derives from their initial discovery in 1967, reflecting their elution fraction (14) and migration position (0.33) during protein purification from bovine brain (Grasso et al., 1977; Moore & McGregor, 1965). Following advances in DNA sequencing during the 1980s, 14-3-3 proteins were increasingly recognized as important regulatory molecules, with over 300 publications now appearing annually. Although early studies focused on their roles in cell cycle regulation and signal transduction (Aitken et al., 1992), recent work has revealed crucial tissue-specific functions, particularly in the regulation of ion channel trafficking and activity. The role of 14-3-3 proteins in regulating cardiac function and excitation–contraction (EC) coupling, graphically illustrated in the Abstract Figure, represents an emerging and exciting area of investigation. In this manuscript we review current knowledge of cardiac 14-3-3 proteins and their role

in regulating heart function, highlighting both established mechanisms and outstanding questions in the field.

## 14-3-3 structure

### Isoforms.

The seven human 14-3-3 proteins are encoded by distinct genes: $\beta$ (YWHAB, chromosome 20), $\epsilon$ (YWHAE, chromosome 17), $\gamma$ (YWHAG, chromosome 7), $\eta$ (YWHAH, chromosome 22), $\tau$ (YWHAQ, chromosome 2; sometimes referred to as 14-3-3$\theta$), $\zeta$ (YWHAZ, chromosome 8) and $\sigma$ (SFN, chromosome 1). Despite their dispersal across different chromosomes, these isoforms maintain remarkable sequence conservation, sharing 59%–87% amino acid identity between pairs (Fig. 1). The evolutionary pressure to maintain isoform-specific functions is highlighted by the observation that some isoforms share greater sequence identity with their orthologs in distant species than with other human isoforms. For example human 14-3-3$\epsilon$ shows 84% sequence identity with its Drosophila ortholog but only 69% with human 14-3-3$\zeta$, suggesting early differentiation and functional specialization of individual isoforms during evolution (Wang & Shakes, 1996).

14-3-3 proteins are between 245 and 255 amino acids and have predicted molecular masses of 27.7–29 kDa. The structures of all human 14-3-3 isoforms have been solved for nearly 20 years and reviewed many times (recent review (Obsilova & Obsil, 2022)). Given the extensive structural analyses available, we will keep our structural detail brief and focus instead on key features that underlie 14-3-3's functional roles in the heart. Each 14-3-3 monomer adopts a highly conserved structure comprising nine $\alpha$-helices that form an amphipathic groove, with a more variable disordered C-terminal tail. The binding groove contains three key positively charged amino acids (Arg-57, Arg-130 and Tyr-131; numbering here refers to the $\epsilon$ isoform) that form a basic pocket essential for recognition of phosphorylated target protein binding motif. The groove's structural flexibility allows it to accommodate diverse phosphorylated targets with multiple distinct binding motifs, and in some cases non-phosphorylated sequences that do not fully match the consensus sequences (Yang et al., 2006). This architectural design enables 14-3-3 proteins to serve as versatile molecular scaffolds, capable of binding hundreds of different (mostly) phosphoproteins.

### Dimerization.

14-3-3 isoforms spontaneously dimerize (Aitken et al., 2002; Chaudhri et al., 2003), each elongated 'c'-shaped binding groove joining along one outer edge so that the binding grooves are parallel (but in opposite directions) to each other, and the whole structure forms a rounded 'w' with the dimer interface in the centre when viewed from above. Most isoforms strongly prefer the dimer state and function only as a dimer (Chaudhri et al., 2003; Yang et al., 2006). There have been reports that some isoforms can regulate certain targets as a monomer (Zhou et al., 2003); however, it is unclear if this occurs in physiological conditions due to reduced stability of the monomeric form (Messaritou et al., 2010; Sluchanko & Gusev, 2012; Yang et al., 2006). 14-3-3 generally dimerizes via two conserved salt bridges between Arg18–Glu 89 and Asp 21–Lys 85 (numbering here refers to the 14-3-3$\zeta$ isoform) (Gardino et al., 2006). However in this portion of the

structure the isoforms are somewhat less conserved; 14-3-3$\epsilon$ does not have the second salt bridge; 14-3-3$\zeta$, $\beta$ and $\tau$ have an additional salt bridge between Glu 5 and Lys 74; and 14-3-3$\sigma$ has a unique salt bridge between Lys 9 and Glu 83 (Gardino et al., 2006). These structural differences at the dimer interface create distinct dimerization preferences among isoforms. For example 14-3-3$\sigma$ predominantly forms homodimers due to unique stabilizing modifications along its dimer interface, whereas 14-3-3$\epsilon$ preferentially forms heterodimers with other 14-3-3 isoforms (Chaudhri et al., 2003; Gardino et al., 2006; Yang et al., 2006). This combination of isoform-specific dimerization preferences and tissue-specific expression patterns provides an additional layer of regulation over target binding specificity.

## 14-3-3 regulation

### Binding motifs.

14-3-3 proteins recognize three conserved binding motifs on target proteins: mode I (-RSx($_p$S/T)xP-), mode II (-Rx$\phi$x($_p$S/T)xP-) and mode III ($_p$S/Tx$_{1-2}$-COOH), with x representing any residue, $\phi$ representing any aliphatic or aromatic residue and $_p$S/T indicating a phosphorylated serine or threonine (Coblitz et al., 2006; Ganguly et al., 2005; Yaffe et al., 1997). Although modes I and II engage the full binding pocket, mode III binding is unique in that it binds to the C-terminal tail of the target protein and occupies only the first half of the binding pocket (Ganguly et al., 2005; Wurtele et al., 2003). These binding motifs frequently overlap with kinase recognition sequences, creating a direct mechanistic link between phosphorylation and 14-3-3-dependent regulation. Although the three binding modes are conserved and relatively specific, actual sequences of target proteins can vary with binding affinity generally correlating with how closely the target sequence matches the conserved motif (Johnson et al., 2010; Tinti et al., 2014). Notably phosphorylation-independent binding can occur when targets contain phosphomimetic residues such as Asp or Glu (Petosa et al., 1998; Wang et al., 1999). The binding motifs between isoforms are nearly identical, but many target proteins show preference or even exclusive binding towards particular isoforms (Gogl et al., 2021). Much of the isoform specificity is thought to come from steric hindrance of the differential outer structure with the target protein and subtle differences in the interacting residues in the binding pockets of different 14-3-3 isoforms (Gardino et al., 2006; Yaffe et al., 1997).

The tendency of 14-3-3 to dimerize adds additional complexity to its regulation. Proteins with two 14-3-3 binding sites show increased 14-3-3 binding stability (Yaffe et al., 1997), which lends itself to several types of complex regulation. Target proteins with two lower-affinity sites can bind 14-3-3 at once and stably form what would otherwise be an unstable interaction. In this way 14-3-3 can act as a coincidence detector if both sites are phosphorylated by different kinases, as 14-3-3 will be unable to bind to either site alone. This feature can also provide redundancy if both sites can bind 14-3-3 alone and have high enough affinity to stay bound individually or together. This is important for fast sequestration of potentially harmful signalling molecules, such as the apoptosis-inducing protein BAD, which has two near-perfect 14-3-3 consensus sites and upon phosphorylation is immediately bound by 14-3-3 and sequestered (Yaffe et al., 1997; Zha et al., 1996).

On the contrary having paired sites can make 14-3-3 binding more specific if these paired sites require binding of specific pairs of isoforms, as with the epithelial sodium channel (ENaC) in the kidney which binds only to 14-3-3$\beta$-$\epsilon$ dimers (Liang et al., 2008) or phosphatase slingshot which binds 14-3-3$\zeta$-$\tau$ dimers (Kligys et al., 2009). A final implication of these paired sites is the ability to link proteins together. Although this will not be possible for many proteins due to steric hinderances in the relatively small distance between the binding sites (Gardino et al., 2006), it has been shown in several cases between subunits or monomers of the same protein (Horvath et al., 2021; Ottmann et al., 2007; Rajakulendran et al., 2009; Wurtele et al., 2003) or between different proteins (Braselmann & McCormick, 1995; van der Hoeven et al., 2000).

### Phosphorylation.

One of the best-studied forms of 14-3-3 regulation is phosphorylation of 14-3-3 itself. Although most studies examining 14-3-3 phosphorylation have been conducted in non-cardiac tissues, the high degree of conservation of these regulatory sites suggests similar mechanisms may operate in the heart. Most of the studies on phosphorylation of 14-3-3 have been done on the 14-3-3$\zeta$ isoform at S58, S184 and T232. Some of these findings can be extrapolated to other isoforms, but in many cases these sites are in less conserved regions. The S58 site, which lies in the dimer interface, has many known kinase motifs, such as sphingosine-dependent kinase 1 (SDK1) (Megidish et al., 1998), protein kinase B (PKB/akt) (Powell et al., 2002), MAP kinase-activated protein kinase 2 (MAPKAPK2) (Powell et al., 2003), PKA (Gu et al., 2006), oxidative stress-response kinase 1 (SOK-1) (Zhou et al., 2009) and protein kinase C $\delta$ (PKC$\delta$) (Gerst et al., 2015; Toker et al., 1992). Phosphorylation at this site disrupts 14-3-3$\zeta$ dimerization (Gu et al., 2006; Trosanova et al., 2022), which impairs its ability to interact with most target proteins, as the dimeric state is typically required for effective binding (Kanno & Nishizaki, 2011; Powell et al., 2003; Woodcock et al., 2003).

Because this serine and most of the surrounding amino acids are found in nearly all human 14-3-3 isoforms aside from 14-3-3$\tau$ and $\sigma$, we may expect this regulation to carry across to other isoforms. Indeed SDK1 (a cleaved section of PKC$\delta$ (Hamaguchi et al., 2003)) has been shown to phosphorylate 14-3-3$\beta$, $\eta$ and $\zeta$, but not 14-3-3$\sigma$ or $\tau$ (14-3-3$\epsilon$ and $\gamma$ were not tested) (Megidish et al., 1998). Increased phosphorylation of 14-3-3$\beta$, $\eta$, $\zeta$, $\gamma$ and $\tau$ has been shown following P21 activated kinase 6 (PAK6) activation, but the phosphorylation site was only confirmed for 14-3-3$\gamma$, so this is not conclusive for the other isoforms (Civiero et al., 2017). Full-length PKC has also been shown to phosphorylate 14-3-3$\beta$, $\eta$, $\zeta$ and $\gamma$, but not 14-3-3$\sigma$, $\tau$ or $\epsilon$ (Toker et al., 1992). Although this generally corresponds to the sequences, where 14-3-3$\beta$, $\eta$, $\zeta$ and $\gamma$ have the full conserved PKC consensus site, and 14-3-3$\sigma$ and $\tau$ lack the phosphorylatable serine, the lack of 14-3-3$\epsilon$ phosphorylation is interesting, as the main difference between 14-3-3$\zeta$ and $\epsilon$ is the lack of a serine immediately before S58, which is not necessarily part of the consensus sequence. However this could be due to availability and steric differences, as 14-3-3$\epsilon$ dimerization is somewhat different from other isoforms to form stable dimers with its missing salt bridge (Gardino et al., 2006).

Phosphorylation at the second site on 14-3-3$\zeta$, Ser184, has a more variable effect. It strengthens interactions with PKC (Aitken, Howell, Jones, Madrazo, Martin et al., 1995; Aitken, Howell, Jones, Madrazo, & Patel, 1995) and with heat shock protein B6 (HspB6) but not tau protein (Sluchanko et al., 2011), and apoptosis signalling molecule Bax shows a decrease in association following phosphorylation by c-Jun NH2-terminal kinase (JNK) (Tsuruta et al., 2004), suggesting that this enhancement may be target specific. This site is commonly phosphorylated in the 14-3-3$\beta$ and $\zeta$ isoforms, leading to the separate naming of the phosphorylated and unphosphorylated forms when they were first named (called 14-3-3$\alpha$ and 14-3-3$\delta$, respectively) (Aitken, Howell, Jones, Madrazo, & Patel, 1995). 14-3-3$\epsilon$ and 14-3-3$\sigma$ also have serines at the same site, and although 14-3-3$\sigma$ has also been shown to be phosphorylated by JNK (Tsuruta et al., 2004), it does not seem as common *in vivo*, and there have been no definitive studies showing S184 phosphorylation for 14-3-3$\epsilon$.

The final site on 14-3-3$\zeta$, T232, is found on the disordered C-terminal tail. Phosphorylation of this site by casein kinase I (CKI) inhibits binding to targets (Dubois et al., 1997), potentially by stabilizing a disordered region that can occupy the binding site as rest and must be moved in order for a target to bind (Obsilova et al., 2004). The same position on 14-3-3$\tau$ is a serine, which can also be phosphorylated by CKI to the same effect (Dubois et al., 1997). Otherwise no other isoform shares a phosphorylatable residue in this position.

Outside of the 14-3-3$\zeta$ isoform there have been studies investigating phosphorylation of 14-3-3$\beta$ and 14-3-3$\tau$ isoforms at S130, which is found only on 14-3-3$\beta$, and T141, which is found on both. These isoforms facilitate complex formation between PKC-$\zeta$ and Raf-1 and are then phosphorylated by the PKC-$\zeta$ to break up the complex (van der Hoeven et al., 2000). In the future it will be important to investigate how this regulation translates to cardiac targets, particularly given the nano-domains and compartmentalization that are so crucial to kinase regulation in cardiac myocytes (Fu et al., 2013).

### 14-3-3 expression.

Although a recent review summarized the current knowledge of human 14-3-3 protein isoform distribution across multiple tissues, the heart was notably excluded from this analysis (Sengupta et al., 2020). Here we highlight the present understanding of cardiac 14-3-3 isoform expression, which exhibits some species-specific variations. In rodents RT-PCR analysis of rat cardiac tissue revealed an mRNA abundance hierarchy of YWHAE>YWHAH>YWHAZ>YWHAB>YWHAG>YWHAQ that held when comparing atrial with ventricular samples (Arakel et al., 2014). This pattern largely mirrors the hierarchy of mouse heart protein isoform expression ($\epsilon$>$\gamma$>$\zeta$>$\beta$>$\eta$>$\tau$), particularly at the extremes of abundance as quantified through SILAC proteome analysis (Qu et al., 2022). Human expression profiles, however, show inconsistencies across studies. Analysis of human transcriptome data (BioProject: PRJNA264807; GSE62689) by Qu and colleagues identified an abundance pattern of YWHAG>YWHAE>YWHAB>YWHAH>YWHAZ>YWHAQ, whereas Thompson and Goldspink, using the Human Protein Atlas consensus dataset (www.proteinatlas.org), reported a different expression hierarchy: YWHAQ>YWHAE>YWHAB> YWHAG>YWHAZ>YWHAH (Thompson & Goldspink,

2022). The variations observed in human samples may reflect not only methodological differences but also genuine individual-to-individual variations in isoform expression, potentially influenced by factors such as age, sex, genetic background and underlying health conditions. A systematic analysis accounting for these variables could help establish more definitive expression patterns and potentially identify clinically relevant expression signatures.

Given 14-3-3's extensive interactome and crucial role in cell survival, it is logical that its expression is tightly regulated in a tissue- and isoform-specific manner. 14-3-3 protein and RNA levels vary with different disease conditions and during development, for example 14-3-3$\epsilon$ protein peaks during days 14.5 to 16.5 of rat heart development (Luk et al., 1998), and after acute myocardial injury 14-3-3$\gamma$ and $\eta$ mRNA rapidly increase, accompanied by an increase in general 14-3-3 protein levels (He et al., 2006). On the contrary, mRNA expression of 14-3-3$\zeta$ has been shown to be stable enough to use as a reference gene for qPCR in healthy *versus* diseased heart ventricle samples (Molina et al., 2018).

Several transcriptional regulators of cardiac 14-3-3 expression have been identified. Expression of 14-3-3$\epsilon$ has been shown to be controlled by peroxisome proliferator-activated receptor $\delta$ (PPAR$\delta$) and CCAAT/enhancer binding protein-$\beta$ (C/EBP$\beta$) (Brunelli et al., 2007). The clinical relevance of this regulation is highlighted by a case where a mutation reducing C/EBP$\beta$ binding affinity was associated with ventricular non-compaction in a Japanese patient (Chang et al., 2013). In H9c2 cells sirtuin SIRT2 negatively regulates 14-3-3$\zeta$ expression, affecting BAD sequestration and cellular responses to anoxia/reoxygenation injury (Lynn et al., 2008). Despite these insights our understanding of cardiac 14-3-3 expression regulation remains limited. This highlights the difficultly in studying generalized regulation of such a diverse and specialized protein; there are differences between isoforms and tissues, so the likelihood that information about regulation of a single isoform in a particular tissue is unlikely to translate widely.

### Inhibitors and activators.

Recent comprehensive reviews of 14-3-3 stabilizers and inhibitors are available (Somsen et al., 2024; Stevers et al., 2018). We will focus here on the key inhibitors and activators that have proven particularly useful for investigating cardiac 14-3-3 function. The fungal toxin fusicoccin A is a small molecule that fits into half of the 14-3-3 binding groove and dramatically stabilizes binding between 14-3-3 and targets that utilize mode III binding by filling the other half of the binding site (Wurtele et al., 2003). Fusicoccin A has limited use as a general 14-3-3 stabilizer, as most 14-3-3 targets do not engage in mode III binding, but it is very useful for known mode III targets and to differentiate mode I and II binding proteins from mode III, as it increases the stability so greatly. Interestingly although fusicoccin A generally stabilizes all mode III interactions regardless of the isoform, some $\sigma$ targets show an additional increase in stability over other isoforms or even other mode III $\sigma$ target proteins (Sengupta et al., 2020).

BV02 is a small cell-permeable molecule that is structurally similar to the core structure of the 14-3-3 binding motif and acts as a competitive inhibitor by occupying the binding groove (Iralde-Lorente et al., 2019). The main advantage of this inhibitor is that it is cell permeable

and does not require transfection, meaning it can be used easily in myocytes and at relatively short time points. It is completely non-selective and will indiscriminately inhibit all 14-3-3 isoforms (Stevers et al., 2018), which can lead to apoptosis depending on dose and time.

R18 is a small protein that contains a sequence that mimics a phosphorylated 14-3-3 mode I or II binding motif, which allows it to also function as a competitive inhibitor (Petosa et al., 1998; Wang et al., 1999). A dimerized version of R18 called difopein (*di*meric *fo*urteen-three-three *pe*ptide *in*hibitor) can be used to bind to both sides of 14-3-3 at once for increased stability (Masters & Fu, 2001). Like BV02 this inhibitor is non-selective and can also lead to apoptosis. The protein does not pass through the membrane and must be transfected (in plasmid form) or injected into the mouse or cell, but it does have the advantage of optionally carrying a fluorescent tag, so expression can be verified. In addition a non-binding mutated version of the same overall structure is available to serve as a control. Transgenic mice expressing difopein under a neuronal promoter have previously been used as a tissue-specific 'functional knockout' to assess the effects of simultaneous reduction in activity of all 14-3-3 isoforms (Qiao et al., 2014).

### Knockout and knockdown.

The study of individual 14-3-3 isoforms is complicated by their overlapping target specificities, as isoforms can often functionally compensate for each other depending on their relative binding affinities (Acevedo et al., 2007; Gogl et al., 2021). Further complicating genetic approaches, germline deletion of many 14-3-3 isoforms result in embryonic lethality. For example global 14-3-3$\tau$ knockout proved embryonic lethal in mice, and although heterozygous mice survived and had apparently functionally normal hearts, they displayed increased susceptibility to apoptosis and pathological remodelling (Lau et al., 2007). Global deletion of 14-3-3$\epsilon$ typically results in perinatal lethality, with major neuronal migration defects (Toyo-oka et al., 2003) and stunted cardiac development resulting in ventricular non-compaction (Gittenberger-de Groot et al., 2016; Kosaka et al., 2012). The severity of this phenotype is influenced by genetic background, with inbred strains showing complete lethality and mixed background mice showing partial survival (Wachi et al., 2017). A similar pattern is observed with 14-3-3$\zeta$ knockouts, where deletion causes embryonic lethality in inbred lines and significant early postnatal mortality even in mixed background strains (Cheah et al., 2012; Yang et al., 2017). The first report of a 14-3-3$\gamma$ knockout found surprisingly normal mice, with no differences from wild-type littermates and no lethality (Steinacker et al., 2005), but a later study using a 14-3-3$\gamma$ knockout generated using the gene trapping approach (Kurogi et al., 2017) found all homozygous mice died before 21 days old, and the heterozygous mice exhibited growth deficits (Kim et al., 2019). Homozygous knockout of 14-3-3$\eta$ produced morphologically normal mice, with sterile male mice (Buret et al., 2016). To our knowledge 14-3-3$\beta$ or 14-3-3$\sigma$ knockout mice have not yet been reported in the literature.

So far most 14-3-3 knockouts have been global germline knockouts which are initiated based on the protein's endogenous promoter, preventing any developmental roles. Most isoforms of 14-3-3 have clear early developmental roles as evidenced by embryonic or perinatal lethality and smaller heterozygotes, so outside of developmental studies

an inducible knockout may be more effective for future studies. Cre-driven knockouts have so far been successful in avoiding much of the 14-3-3$\epsilon$ knockout perinatal lethality, with heterozygous double knockouts (heterozygous 14-3-3$\zeta$ knockout with cre-driven homozygous 14-3-3$\epsilon$ knockout or homozygous 14-3-3$\zeta$ knockout with cre-driven heterozygous 14-3-3$\epsilon$ knockout) regularly surviving 2 months or longer (Toyo-oka et al., 2014). 14-3-3$\epsilon$ knockout mice alone survived at least long enough to reproduce though fertility of both males and females was affected (Eisa et al., 2019, 2021). Interestingly cre-driven 14-3-3$\eta$ knockout males showed normal fertility (Eisa et al., 2021) in contrast to the gene trap approach (Buret et al., 2016). shRNA-mediated knockdown has also proven valuable for studying 14-3-3 function (Izumi et al., 2014; Yang et al., 2017) and remains a good alternative, offering temporal control while avoiding developmental complications.

### Dominant negative.

Because 14-3-3 generally functions only as a dimer, overexpressing a dimerization-capable but non-target-binding monomer can have a dominant-negative effect on endogenous 14-3-3 (Chang & Rubin, 1997; Zhang, Chen et al., 1999; Zhang, Xing et al., 1999). This approach will likely still run into the same complications of potential compensation, cross-reactivity, heterodimerization and tissue-specific isoform expression levels as knockdowns or knockouts. On the contrary this approach is the only one that allows at least partial separation of 14-3-3 targets that bind exclusively to both sides of the dimer *versus* those that are still capable of binding to half of the 14-3-3, as shown in this study (Zhou et al., 2003). In addition this will likely yield somewhat different responses than a knockdown or heterozygous knockout. Removal of an isoform will shift the dimerization balance between isoforms and probably increase homodimerization of the remaining isoforms, whereas expressing a non-target-binding version of an isoform will preserve the current equilibrium or perhaps shift it the other direction.

This approach has been successfully used in the heart; for example mice with cardiac-specific postnatal expression of dominant-negative 14-3-3$\eta$ were reported to survive for at least 6 months and appeared healthy with normal cardiac function but were more sensitive to pressure overload (Xing et al., 2000) or diabetes-induced cardiomyocyte apoptosis (Gurusamy et al., 2004).

## General 14-3-3 functions

14-3-3 is thought to function via three overall mechanisms: masking, clamping and scaffolding (Smith et al., 2011), as shown in Fig. 2. Through masking 14-3-3 binding can hide a signalling sequence or motif such as an endoplasmic reticulum (ER) retention sequence or a phosphorylated residue (O'Kelly et al., 2002). This affects trafficking of membrane proteins such as $Ca_V2.2$ (Liu et al., 2015) and prevents dephosphorylation of targets such as the small regulatory protein phospholamban (PLB) (Menzel et al., 2020). Clamping refers to holding proteins in particular conformations by leveraging dual binding sites and the relative rigidity of the 14-3-3 structure (Yaffe, 2002). A classic example of this is the interaction between the circadian regulator serotonin *N*-acetyltransferase and 14-3-3, which stabilizes a favourable conformation of the substrate-binding region of the

enzyme (Obsil et al., 2001). Finally scaffolding occurs when 14-3-3 binds to separate proteins, linking them closely together, as mentioned previously. This mode is crucial for the assembly of the plant $H^+$-ATPase from its inactive dimer form into a functional full hexameric channel (Ottmann et al., 2007; Wurtele et al., 2003). These three modes lead to a variety of distinct yet related functions in cardiac tissue.

### Trafficking.

Facilitation of forward trafficking by 14-3-3 is one of the more classical functions of 14-3-3 (for review see (Smith et al., 2011)). The simplest mechanism for this facilitation is through prevention of the binding of coat protein complex I (COPI). Interactions with COPI promote anterograde trafficking and ER/Golgi retention, and many proteins have overlapping or adjacent COPI/14-3-3 binding sites. Initially COPI was thought to exclusively compete with 14-3-3 binding (Heusser et al., 2006; O'Kelly et al., 2002), but more recent evidence suggests that at least in some cases phosphorylation of the target protein alone prevents COPI binding, whereas 14-3-3 potentially enhances trafficking by preventing dephosphorylation to prolong the signal (Arakel et al., 2014; Kilisch et al., 2016). 14-3-3 interactions with COPI occurs for $K_{ATP}$ (Arakel et al., 2014; Heusser et al., 2006) and TASK-1/TASK-3 (Kilisch et al., 2016; O'Kelly et al., 2002) channels, which are expressed in the heart.

Beyond COPI-mediated retention, 14-3-3 proteins also facilitate forward trafficking by overcoming other ER retention signals. A notable example is the RKR motif, an arginine-based ER retention signal found in various membrane proteins. A screen performed to identify signals capable of overcoming this ER retention sequence identified the SWTY motif as restoring surface expression to Kir2.1 channels fused to RKR (Shikano et al., 2005). This forward trafficking mechanism requires phosphorylation-dependent recruitment of 14-3-3, which effectively masks the ER retention signal. SWTY-like motifs have been discovered in several proteins, including the cardiac TASK-1 K+ channel (Shikano et al., 2005), suggesting this may be a widespread regulatory mechanism. For other cardiac membrane proteins like the L-type $Ca^{2+}$ channel $Ca_V1.2$, 14-3-3 has been shown to influence trafficking, though the precise mechanism remains under investigation, with ER retention signal masking representing a plausible mechanism (Spooner et al., 2025).

### Chaperoning.

On a related note 14-3-3 can act as a chaperone to regulate protein folding, sequester proteins or to localize proteins in particular compartments such as the nucleus. This form of regulation is primarily associated with cell cycle control and apoptosis signalling – for example 14-3-3-mediated sequestration of apoptosis factor BAD to prevent its binding with survival factor $BCL-X_L$ (Zha et al., 1996) – but is also found more broadly. An illustrative case involves the reported interactions between 14-3-3, RGK proteins (Rad, Rem, Rem2 and Gem/Kir) and $Ca^{2+}$ channel auxiliary subunits. Evidence from heterologous expression studies suggests that RGK proteins such as Rad and Rem can sequester calcium channel $\beta$ subunits, and that 14-3-3 binding to RGK proteins might interfere with this interaction (Beguin et al., 2006). However it is important to note several caveats: (1) to our knowledge there is no evidence demonstrating nuclear localization of RGK proteins

in native cardiomyocytes under physiological conditions; (2) heterologous expression systems with non-physiological protein stoichiometries may not accurately reflect *in vivo* interactions; and (3) it remains unclear whether RGK-Ca$_V\beta$ interactions are sufficiently strong to disrupt established Ca$_V\beta$-Ca$_V\alpha$ interactions in the physiological context. Although these studies provide important foundational insights into potential regulatory mechanisms, further investigation in cardiac-specific contexts with appropriate stoichiometries is needed to establish whether 14-3-3 proteins indeed modulate Ca$^{2+}$ channel trafficking through this pathway in the heart.

### Modifying activity.

In addition to regulating the localization and delivery of proteins 14-3-3 can bind directly to various proteins and cause a change in their activity. Many of the other ion channels and transporters that 14-3-3 affects in the heart are regulated this way, such as the hERG potassium channel (Choe et al., 2006; Kagan et al., 2002; Krishnan et al., 2012), PMCA 1/3/4 (Linde et al., 2008; Rimessi et al., 2005) and Na$_V$1.5 (Allouis et al., 2006; Clatot et al., 2017).

### Preventing dephosphorylation.

A final general role of 14-3-3 is hiding phosphorylated residues from dephosphorylation by phosphatases, prolonging the signal. In the heart this is a crucial function, given the importance of $\beta$-adrenergic regulation for overall heart function and stress responses. So far this is known to occur for PLB (Menzel et al., 2020), which regulates the activity of sarco/ER ATPase (SERCA) Ca$^{2+}$ pump and the *h*uman *E*ther-à-go-go-*R*elated *G*ene-encoded K$^+$ channel hERG (Kagan et al., 2002). In future studies it will be interesting to see if this is a conserved role for the many PKA targets in the heart.

## 14-3-3 in cardiac health and disease

Mutations in 14-3-3 itself are not considered causative for any cardiac diseases as of yet, and considering its ubiquitous and highly conserved nature any major alterations to its function are likely to be fatal. However 14-3-3 levels, isoform distribution and target binding have all been implicated in various heart diseases and pathological remodelling, so cardiac roles in 14-3-3 regulation remain an important area of study. All 14-3-3 isoforms, with the exception of $\sigma$, are highly expressed in basal conditions in the heart (Thompson & Goldspink, 2022), although there does seem to be some amount of variation between the Human Protein Atlas consensus dataset (Thompson & Goldspink, 2022) and individual reports (Arakel et al., 2014; Qu et al., 2022), suggesting variability that is not yet well understood. 14-3-3 regulation of cardiac targets can be roughly divided into three categories: EC coupling, myocyte development and cardiac stress responses.

### EC coupling.

Cardiac EC coupling refers to the cellular process that occurs in cardiomyocytes, in which an electrical signal, known as an action potential, is transduced into the mechanical contraction of the heart muscle. This action potential originates in the pacemaker cells of the sinoatrial node and propagates as a wave of depolarization throughout the heart. A

specialized conduction system facilitates the precisely choreographed transmission of the action potential through the ventricles, ensuring that the electrical impulse arrives at the desired location at the appropriate time. This precise co-ordination allows the heart to beat efficiently, with the atria contracting first, followed by the ventricles, optimizing the blood pumping action of the heart.

A crucial element of cell-to-cell conduction in myocytes is the gap junctions, which are low-resistance channels that allow cell-to-cell coupling of adjacent cardiomyocytes at their borders, called intercalated disks. Gap junctions allow the myocardium to behave as a single unit, termed a functional syncytium. The most abundant gap junction-forming protein in the heart is connexin 43 (Cx43) (Gutstein et al., 2001; Lillo et al., 2023). Several reports have linked 14-3-3 binding to Cx43, which facilitates its exit from the ER, possibly by occluding a COPI-dependent RxR-type ER retention signal present on the Cx43 C-terminus near serine residues that are potential 14-3-3 binding sites (Batra et al., 2014; Majoul et al., 2009). This interaction allows the subsequent trafficking of Cx43 to the Golgi apparatus and onward to its site of action, which in cardiomyocytes is at the intercalated disks. Immunofluorescence studies have demonstrated 14-3-3 enrichment at intercalated disks in human heart samples (Smyth et al., 2014), isolated rabbit cardiomyocytes (Allouis et al., 2006) and mouse cardiomyocytes (Spooner et al., 2025), suggesting a possible interaction between 14-3-3 and Cx43. Furthermore in a keratinocyte cell line called HaCaT, siRNA-mediated knockdown of 14-3-3$\tau$ was found to induce enlargement of Cx43-containing gap junction plaques implying a further role for 14-3-3 in internalization of Cx43 from the plasma membrane at least in that immortalized cell line (Smyth et al., 2014). Cx43 internalization was also absent when cells were transfected with a Cx43 mutant in which Ser373 was mutated to a non-phosphorylatable alanine residue. These results implied that 14-3-3-binding to phosphorylated Ser373 on Cx43 could destabilize their expression at intercalated disks. Smyth et al. further reported increased appearance of internalized Cx43 phosphorylated at Ser373 and Ser 368 during cardiac ischaemia in mouse hearts. They proposed a model where phosphorylation-dependent interaction of 14-3-3 with Cx43 ultimately leads to their internalization, ubiquitination and degradation, impairing cell-to-cell communication. These studies reveal that 14-3-3 can have multiple effects on the same protein, which may be site-specific or isoform-specific binding.

Beyond the initial spread of the action potential EC coupling in the heart is driven primarily by $Ca^{2+}$; an action potential depolarizes the membrane, then the subsequent $Ca^{2+}$ entry through voltage-gated $Ca_V1.2$ channels activates nearby RyRs, triggering $Ca^{2+}$-induced $Ca^{2+}$ release from the sarcoplasmic reticulum (SR). This combined $Ca^{2+}$ influx generates a sufficient concentration of calcium ions to bind to the troponin complex, which unlocks the myofilaments and allows contraction to occur. $\beta$-adrenergic receptor ($\beta$AR) signalling through PKA affects many targets, generally causing the heart to beat stronger and faster (Bers, 2002; Harvey & Hell, 2013). 14-3-3 regulates many of the ion channels involved in EC coupling, with a general trend in increasing basal contractility and supporting the PKA-mediated increases in inotropy (contractility), lusitropy (relaxation rate) and potentially chronotropy (heart rate) (Abstract Figure). Although no studies have investigated 14-3-3 regulation of the channel that underlies the pacemaking potential of the heart, *h*yperpolarization-activated *c*yclic *n*ucleotide-gated channel 4 (HCN4), recently the

primarily neuronal isoform HCN1, was found to be regulated by 14-3-3$\zeta$ (Lankford et al., 2022). Intriguingly of the two sites found to bind to 14-3-3, one of these sites is conserved in HCN4, leaving the possibility of 14-3-3 regulation of the pacemaker potential as well.

### Calcium handling.

It has been previously established that 14-3-3 regulates the trafficking and inactivation of $Ca_V2.2$ (Li et al., 2006; Liu et al., 2015) and can likely bind to $Ca_V1.2$ as well, although this was not tested in a physiological context (Yang et al., 2013). We have recently published the first conclusive direct link between 14-3-3 and $Ca_V1.2$, showing that 14-3-3$\epsilon$ and $Ca_V1.2$ form complexes, and 14-3-3 is localized alongside $Ca_V1.2$ in the t-tubule sarcolemma. The nanoscale distribution, trafficking and activity of $Ca_V1.2$ in basal and $\beta$-adrenergic-stimulated conditions are regulated by 14-3-3 levels, with the overall effect of enhancing $\beta$-adrenergic signalling (Spooner et al., 2025).

The previously mentioned 14-3-3 regulation of RGK proteins, such as Rad (Beguin et al., 2006), likely contributes to $Ca_V1.2$ activity regulation. Rad inhibits the channel both directly through binding to the auxiliary $Ca_V\beta$ subunit (Katz et al., 2021; Papa et al., 2022) and indirectly by modulating $Ca_V1.2$ trafficking (Beguin et al., 2006; Yada et al., 2007). PKA phosphorylation of Rad at 14-3-3 binding sites relieves this inhibition (Beguin et al., 2006; Katz et al., 2021; Papa et al., 2022). Recent AlphaFold3 predictions suggest 14-3-3$\epsilon$ can bind to the $\alpha_{1C}$ subunit and to multiple phosphorylated components of the $Ca_V1.2$ channel complex, including $Ca_V\beta$ and Rad (Spooner et al., 2025). These structural models raise intriguing questions about whether 14-3-3$\epsilon$ binding to $Ca_V\beta$ and/or Rad might obstruct or competitively inhibit the $Ca_V\beta$-Rad interaction.

The Satin group recently developed an innovative mouse model featuring truncation of Rad's polybasic C-terminal region (Elmore et al., 2024). This positively charged polybasic domain, found in Rad and other RGK proteins, normally facilitates membrane localization through electrostatic interactions with negatively charged phosphatidylinositol lipids (Heo et al., 2006; Correll et al., 2007). The truncation causes Rad to mislocalize to the cytosol instead of its normal t-tubule position, disrupting its interaction with $Ca_V\alpha_{1C}$-associated $Ca_V\beta$ and precluding both its basal suppression of $I_{Ca}$ and its $\beta$-adrenergic signalling-triggered augmentation (Elmore et al., 2024). Notably this truncation removes the S300 14-3-3 binding site on Rad (Finlin & Andres, 1999; Beguin et al., 2006; Spooner et al., 2025). Previous studies have linked 14-3-3 expression to RGK protein redistribution from the nucleus to the cytosol in a mechanism requiring the intact Rad C-terminal (Beguin et al., 2006). These observations, when viewed alongside the recent Satin lab findings, raise the intriguing, though as yet unexplored, possibility that 14-3-3 might contribute to Rad localization in cardiomyocytes. This hypothesis warrants further investigation.

Marx and colleagues further demonstrated that Rad must dissociate from both the membrane (Papa et al., 2024) and from $Ca_V\beta$ to permit $\beta$-adrenergic regulation of $Ca_V1.2$ channels (Liu et al., 2020). This process depends on PKA-mediated phosphorylation of two specific C-terminal serine residues (S272 and S300), which generates electrostatic repulsion from the negatively charged membrane (Yang et al., 2019; Papa et al., 2024). Charge-substitution experiments with aspartic acid residues confirmed this mechanism, showing relief of Rad's

suppressive effect on channel open probability with increasing negative charge (Papa et al., 2024). Notably membrane-anchored Rad (via CAAX fusion) remains locked in its inhibitory state despite phosphorylation or charge substitutions (Papa et al., 2024).

The potential role of 14-3-3 proteins in this system is particularly intriguing as Ser300 is a known 14-3-3 binding site. Although 14-3-3 preferentially binds to phosphorylated serines and threonines, it can also interact with aspartic acid residues (Petosa et al., 1998), suggesting 14-3-3 might contribute to the effects observed with charge-substituted Rad mutants. Competition studies with the related RGK protein Kir/Gem revealed mutually exclusive binding relationships with 14-3-3, calmodulin (CaM) and $Ca_V\beta$, that is Kir/Gem can bind only one of those interaction partners at a time (Beguin et al., 2005). If Rad exhibits similar exclusivity, 14-3-3 binding to phosphorylated or charge-substituted Rad could sequester it from its low-affinity interaction with $Ca_V\beta$ (Xu et al., 2015), contributing to channel disinhibition. Membrane anchoring might sterically block 14-3-3 access to these binding sites, explaining the locked inhibitory state of CAAX-Rad fusion proteins. Understanding these complex interaction networks will be crucial for uncovering the full extent of Rad-dependent and Rad-independent regulation of $Ca_V1.2$ by 14-3-3.

On the other side of the dyad ryanodine receptors (RyR2) serve as crucial SR calcium release channels. To our knowledge no studies have investigated direct or indirect 14-3-3 regulation of RyRs. RyR2s are phosphorylated by PKA and CaMKII, which open the possibility of 14-3-3 regulation, though there have been contradictory reports in the field about the exact effects and role of this phosphorylation in health and disease (Dobrev & Wehrens, 2014).

$Ca^{2+}$ homeostasis is essential for proper cardiomyocyte function. Over the steady state $Ca^{2+}$ influx and efflux must balance as must SR $Ca^{2+}$ release and reuptake (Eisner et al., 2013). Without this equilibrium myocytes would either accumulate $Ca^{2+}$ and be unable to relax or gradually deplete their $Ca^{2+}$ stores and lose contractile force. Of course when regulatory pathways, such as $\beta$-adrenergic signalling, alter $Ca^{2+}$ handling, transient imbalances can occur, leading to changes in SR calcium load before a new steady state is established. SR calcium reuptake is performed by SERCA, a calcium pump that is inhibited by PLB. When PLB is phosphorylated by PKA, it detaches from SERCA, relieving the inhibition. 14-3-3 prolongs the phosphorylated state of PLB (Menzel et al., 2020), shielding it from dephosphorylation. This protection mechanism prolongs SERCA's enhanced activity, resulting in both accelerated relaxation and augmented contractility through increased SR calcium stores and subsequent heightened calcium release during EC coupling.

14-3-3 also plays a role in the trans-sarcolemmal extrusion of $Ca^{2+}$ by inhibiting $Na^+/Ca^{2+}$ exchanger protein NCX (Pulina et al., 2006). NCX1–3 was all shown capable of pulling down 14-3-3$\beta$, $\zeta$, $\tau$ and $\epsilon$ (no other isoforms appeared to be tested), though the initial yeast two-hybrid screening that identified the interaction found only 14-3-3$\zeta$ and $\epsilon$. Overexpression of 14-3-3$\epsilon$ reduced the ability of NCX to clear calcium from the cytosol without a decrease in the surface expression, suggesting a direct effect on NCX activity rather than a trafficking function. NCX proteins are known to be phosphorylated, and 14-3-3 affinity for NCX was markedly increased in phosphorylation conditions *versus* in

unphosphorylated conditions, although the specific consensus site and kinase were not identified in this study (Pulina et al., 2006).

14-3-3 also inhibits the PMCA isoforms expressed in the heart, PMCA1 (Linde et al., 2008) and PMCA4 (Rimessi et al., 2005). Interestingly this interaction does not seem to be PKA or PKC dependent as the N-terminal segment of the pump used as bait to test for 14-3-3 interactions contains a 14-3-3 consensus site (RLKTSP) without a PKA or PKC consensus motif, though PMCA pumps do have well-described PKA and PKC sites elsewhere (Rimessi et al., 2005; Linde et al., 2008). If 14-3-3 is binding to this identified consensus site, it is unlikely to be non-phosphorylation dependent, given that the 14-3-3 targets that are known to bind in a non-phosphorylation dependent manner generally have a phosphomimic such as aspartic acid in place of the serine or threonine (Petosa et al., 1998). This invites speculation about the potential of an additional unknown phosphorylation site or a phosphorylation-independent 14-3-3 binding site elsewhere on the N-terminal segment.

### Potassium channels.

Potassium channels maintain the resting membrane potential and repolarize the cell following an action potential. The hERG potassium channel is susceptible to blockade by numerous pharmaceutical compounds, potentially inducing long QT syndrome as an adverse effect and impairing cardiac repolarization (Mitcheson, 2008). 14-3-3$\epsilon$ and $\eta$ (other isoforms do not appear to have been tested) bind to two PKA consensus sites on the hERG potassium channel – though 14-3-3 can bind to either site alone, binding to both sites as a dimer is required for changes to hERG activity (Kagan et al., 2002). PKA-dependent binding of 14-3-3 increases hERG current density and accelerates its activation, while prolonging its phosphorylation to repolarize the cell more quickly under $\beta$-adrenergic signalling (Kagan et al., 2002). Although hERG protein abundance increases during $\beta$-adrenergic stimulation, 14-3-3 has not been linked to its trafficking (Krishnan et al., 2012).

As previously described in the trafficking section, 14-3-3 regulates TASK-1 and TASK-3 (O'Kelly et al., 2002; Kilisch et al., 2016), which are involved in maintaining the action potential and contributing to repolarization, in addition to their involvement in cardiac remodelling (Duan et al., 2020). TASK-1 has a lower 14-3-3 affinity than TASK-3 due to an additional serine close to the central 14-3-3 binding serine that blocks 14-3-3 binding when phosphorylated, and so its regulation is thought to be more intricate (Kilisch et al., 2016). As a consequence we would expect TASK-3 to be much more dependent on 14-3-3 regulation than TASK-1 in high phosphorylation conditions. Considering that loss of TASK-1, but not loss of TASK-3, was protective for pathological remodelling (Duan et al., 2020), this differential regulation may represent an adaptive stress response. The release of TASK-1 and TASK-3 from the ER/Golgi is PKA phosphorylation dependent, suggesting the cell may rely on pre-synthesized pools of channels that can be mobilized to the sarcolemma when needed (Kilisch et al., 2016). These findings align with previous studies demonstrating that pre-synthesized pools of $Ca_V1.2$ channels are maintained in endosomal reservoirs, facilitating rapid insertion into the t-tubule membrane following $\beta$AR stimulation (Ito et al., 2019; Del Villar et al., 2021; Westhoff et al., 2024), suggesting the possibility that an analogous mechanism may function for $Ca_V1.2$ (Spooner et al., 2025).

14-3-3 also regulates the trafficking of $K_{ATP}$ (Heusser et al., 2006; Arakel et al., 2014). Similarly to TASK-1 and TASK-3, $K_{ATP}$ surface expression is controlled in part via interactions with COPI in the Golgi, which additionally serves as an assembly checkpoint for the full channel, a heterooligomeric complex of four potassium channel subunits (Kir 6.1 or Kir6.2) and four sulfonylurea receptor subunits (SUR1 or SUR2) (Heusser et al., 2006). SUR2-containing channels seem to be less dependent on 14-3-3 regulation and show generally higher expression in ventricular myocytes, whereas SUR1-containing channels have higher expression in the atria (Nichols et al., 2013). Arakel et al. found SUR1-containing channels trafficked to the membrane readily in physiological conditions in the atria, whereas the relatively less-abundant SUR1-containing channels in the ventricles required sustained $\beta$AR activation to be trafficked (Arakel et al., 2014). This insertion of potentially more-active SUR1-containing $K_{ATP}$ channels is thought to be protective in mitigating the harmful effects of prolonged sustained $\beta$AR activation by reducing the activity of the cells affected (Nichols et al., 2013; Arakel et al., 2014).

**Sodium channels.**

Finally 14-3-3 has been shown to regulate $Na_V1.5$, the main driver of the rapid depolarization phase of the ventricular myocyte action potential, though the details of this regulation are not fully clear. The first study linking 14-3-3 $\eta$ with $Na_V1.5$ showed complex formation between the two proteins that resulted in a slower recovery from inactivation, which was thought to be anti-arrhythmogenic (Allouis et al., 2006). In this study no change to overall current density was observed with dominant-negative 14-3-3 $\eta$ or overexpressed 14-3-3 $\eta$, leading the authors to conclude that there was no effect on trafficking. A recent study confirmed the effects of 14-3-3 $\eta$, and also found 14-3-3 $\epsilon$ reduces $Na_V1.5$ expression levels by regulating an essential transcription factor without directly interacting with $Na_V1.5$ (Hu et al., 2024). In this case 14-3-3 $\epsilon$ affected only the current density (due to changes in the number of channels) and no other kinetic properties (Hu et al., 2024).

In addition 14-3-3 has been shown to functionally couple sodium channels together into cooperatively gating dimers (Clatot et al., 2017; Zheng & Deschenes, 2023). Biochemical approaches show that truncated $Na_V1.5$ channels that no longer pull down 14-3-3 can still pull down the full-length channel despite apparently no longer being bound to 14-3-3 (Clatot et al., 2017; Iamshanova et al., 2024); however dominant-negative $Na_V1.5$ mutations have reduced or no dominant-negative effect without 14-3-3 (Clatot et al., 2017). Both of these studies reported adding difopein did not affect inactivation kinetics or current density (Clatot et al., 2017; Iamshanova et al., 2024), contrary to what we would expect given that difopein should inhibit the previously described 14-3-3 $\eta$ and $\epsilon$ effects. This may be due to differences in other regulatory factors across cell types as Hu et al. suggest (Hu et al., 2024), or potentially in relative isoform levels and affinities given that the 14-3-3 $\eta$ and $\epsilon$ effects were more specifically assessed using dominant-negative or siRNA-mediated knockdown (Allouis et al., 2006; Hu et al., 2024). This observation warrants further investigation.

Expanding our understanding of 14-3-3's regulation of cardiac sodium channels, another 2017 study demonstrated that a dominant-negative 14-3-3 $\eta$ mutant (R56, 60A) selectively affected $Na_V1.5$ currents in CHO cells only when co-expressed with Kir2.1, but it had no

effect when either channel was expressed individually (Utrilla et al., 2017). This finding suggests that these adaptor proteins may orchestrate the co-ordinated activity of different ion channel types that work in concert to shape the cardiac action potential.

### Myocyte development.

As previously mentioned, 14-3-3$\epsilon$ levels rise during development in rats, peaking around day 15 and then returning to basal levels (Luk et al., 1998). Furthermore the role of 14-3-3 in cardiac development has been clearly demonstrated by the 14-3-3$\tau$ and $\epsilon$ germline knockout mice, both of which died in utero or shortly after birth with significant cardiac complications even in the heterozygotes (Lau et al., 2007; Kosaka et al., 2012; Gittenberger-de Groot et al., 2016). The 14-3-3$\epsilon$ knockout mice showed left ventricular non-compaction in addition to valve and coronary vessel defects (Kosaka et al., 2012; Gittenberger-de Groot et al., 2016). A human patent with a mutation in the promoter for 14-3-3$\epsilon$ that decreased the affinity for the enhancer and subsequently decreased 14-3-3$\epsilon$ protein levels displayed left ventricle non-compaction, similar to the mouse models (Chang et al., 2013). Recently 14-3-3 proteins have been found to be necessary for the transition of fibroblasts to induced cardiomyocytes by co-ordinating the actions of many transcription factors and effector proteins that may also be involved in cardiac development (Liu et al., 2024).

### Cardiac stress and pathological remodelling.

Many of the studies on cardiac 14-3-3 regulation have been focused on different forms of stress and the tissue's response to them. In most cases 14-3-3 has been shown to be cardioprotective against stress and injury and is often upregulated in response to a stressor. After acute burn injuries when the heart is stressed by a combination of lost plasma and immune activation, cardiac 14-3-3$\gamma$ and $\eta$ mRNA rapidly increase in mice, accompanied by an increase in general 14-3-3 protein levels (He et al., 2006). During brief non-lethal periods of anoxia in mice, cardiac 14-3-3 protein levels rise, correlating with the protective effects of such anoxic conditioning on later episodes of ischaemia (Chen et al., 2007). Similarly elevated 14-3-3$\tau$ alongside elevated apoptosis was found in pups whose mothers had experienced chronic hypoxia (Bae et al., 2003). In immortalized human cardiomyocytes 14-3-3 $\beta$, $\epsilon$, $\gamma$, $\tau$, $\zeta$ and $\eta$ levels increased following endothelin-1 or angiotensin II-stimulated hypertrophy, with $\zeta$ exhibiting the largest fold change from baseline (Mahmud et al., 2023). This increase corresponded to increased 14-3-3 $\zeta$ levels in human patients during acute myocardial infarction (MI) (Mahmud et al., 2023). 14-3-3$\sigma$ has historically been thought to be the only isoform not expressed ubiquitously, as it was not originally detected in tissues other than epithelial cells (Prasad et al., 1992; Leffers et al., 1993). In the heart 14-3-3$\sigma$ is not found under normal *in vivo* conditions, but in stress states such as during taurine deficiency or diabetes mellitus, both of which are associated with the development of cardiomyopathy, its cardiac expression has been shown to be elevated, likely in connection with increased DNA damage (Golubnitschaja et al., 2003; Golubnitschaja, Moenkemann et al., 2006). 14-3-3$\sigma$ is also elevated in stenotic but non-calcified aortic valves, supporting its role in cardiac stress responses (Golubnitschaja, Yeghiazaryan et al., 2006).

Depletion of 14-3-3 prior to an injury generally leads to worse outcomes, though often the mice are not obviously affected by the loss of the 14-3-3 alone. As mentioned previously, heterozygous 14-3-3$\tau$ germline knockout mice seemed functionally normal but were more sensitive to pathological remodelling following MI (Lau et al., 2007). Dominant-negative 14-3-3$\eta$ approaches in models of MI (Sreedhar et al., 2016), pressure overload (Xing et al., 2000) or diabetes-induced cardiomyopathy (Gurusamy et al., 2004; Gurusamy et al., 2006; Sari et al., 2010) all showed increased sensitivity to the stressor and worse outcomes. 14-3-3$\eta$ has been shown to be protective against hyperthyroidism-induced hypertrophy and mitophagy, while knocking down 14-3-3$\eta$ was not sufficient to induce hypertrophy in the absence of hyperthyroidism (Cui et al., 2024).

Many of these stress responses appear to stem from the nuclear factor of activated T cells 3 (NFAT3) hypertrophy pathway, which 14-3-3 regulates on multiple levels. First phosphoinositide 3 kinase (PI3K) recruits and phosphorylates Akt/PKB, activating it. The activated kinase then phosphorylates glycogen synthase kinase-3$\beta$ (GSK3$\beta$), inactivating it, which then decreases NFAT3 phosphorylation levels. Non-phosphorylated NFAT3 is free to move into the nucleus and initiate hypertrophic signalling. NFAT3 can also be phosphorylated by p38, which is in turn regulated by MAPKs such as apoptosis signal-regulating kinase 1 (ASK1/MAP3K5). 14-3-3 regulates the pathway somewhere upstream of PI3K, as non-specific inhibition of 14-3-3 activated PI3K (Du et al., 2005; Liao et al., 2005). In the dominant-negative 14-3-3$\eta$ diabetic cardiac myopathy model GSK3$\beta$ was less active in the 14-3-3-deficient group, correlating with the development of hypertrophy (Gurusamy et al., 2006). Interestingly in this model GSK3$\beta$ and p38 activity were both increased compared to wild-type 14-3-3 controls early in the disease induction, leading to primarily increased apoptosis signalling, suggesting a time-dependent change in the pattern of regulation that causes a shift from apoptosis to hypertrophy (Gurusamy et al., 2004; Gurusamy et al., 2006). On the other arm of the pathway 14-3-3 binds to ASK1/MAP3K5 when it is phosphorylated, inactivating it and promoting p38 activity (Liu et al., 2006). Furthermore 14-3-3 interacts with NFAT3 once it is phosphorylated directly, mediating its cytoplasmic accumulation (Chow & Davis, 2000).

A few stressors have been shown to cause a reduction in 14-3-3 levels rather than an increase. Application of inflammatory cytokine high-mobility group box 1 (HMGB1) to neonatal cardiomyocytes induced a reduction in 14-3-3$\eta$ protein levels without a drop in overall 14-3-3 protein levels accompanied by translocation of NFAT3 from the cytoplasm to the nucleus and hypertrophy (Su et al., 2021). 14-3-3 protein levels were also reduced in rat hearts after development of diet-induced obesity, whereas physical exercise after the development of obesity increased 14-3-3 levels (Pieri et al., 2014).

Finally 14-3-3 activation has been detrimental to cardiac stress responses in some cases. In addition to facilitating forward trafficking of Cx43 (Majoul et al., 2009; Batra et al., 2014), 14-3-3 plays a role in anterograde trafficking and internalization of Cx43 during acute ischaemia (Smyth et al., 2014). This gap junction remodelling is thought to be a protective mechanism for limiting the spread of toxins but is also pro-arrhythmic as it creates zones of altered conductivity. 14-3-3 also binds to and activates the sodium hydrogen exchanger (NHE1), which under normal conditions is crucial for regulating intracellular pH, but during

ischaemia/reperfusion injury leads to increased infarct size and decreases cardiac function (Maekawa et al., 2006). Decreasing its activity experimentally during ischaemia has been beneficial, but clinical attempts have largely been unsuccessful, potentially because the physiological function of NHE1 outside of ischaemia is also important to cardiac health. Maekawa et al. suggest modulation of 14-3-3 binding sites on NHE1 or the kinase that allows its binding, p90 ribosomal S6 kinase (RSK), may be more effective in treating ischaemia without compromising its physiological function (Maekawa et al., 2006).

14-3-3 proteins have long been known to bind histone 3 and histone deacetyltransferases to regulate gene expression (for review see Healy et al. (2011)). During aortic constriction 14-3-3 binds histone 3 following CaMKII phosphorylation to allow initiation of fetal and hypertrophic genes such as Mef2, which generally promote muscle proliferation (Awad et al., 2015). On the other end of the pathway class II histone deacetylases (HDACs) suppress hypertrophic signalling by preventing transcription of hypertrophic genes such as Mef2, whereas 14-3-3 binding to HDACs sequesters them in the cytosol, preventing their suppressive role and promoting hypertrophy (Zhang et al., 2002). Although 14-3-3 activation of both of these pathways resulting in the activation of proliferative genes is consistent with the role of 14-3-3 in promoting cardiac development, it is obviously not optimal during pathological remodelling. PKA-dependent inhibition of PKD was shown to prevent phosphorylation of 14-3-3 binding sites on HDAC5, inhibiting the development of hypertrophy (He et al., 2020). This suggests modulation of the phosphorylation state of the 14-3-3 targets may be more physiologically relevant in this pathway than changing the 14-3-3 levels themselves.

## Conclusions and future perspectives

The work that has been done thus far has shown definitive roles for 14-3-3 in regulating cardiac development, health and EC coupling. Many molecular tools and mouse models have been utilized, and with the increasing popularity of inducible and tissue-specific knockouts, future studies will have the ability to ask increasingly precise questions about how 14-3-3 is regulated and how it controls its various targets. Though much is known about 14-3-3 signalling pathways, many details remain unknown. 14-3-3 regulates hypertrophic gene signalling at several levels and yet is regulated by some of the proteins in the cascades, such as PKB/akt. It regulates the function of many kinases, but its activity is also kinase dependent, both directly through 14-3-3 phosphorylation and indirectly through phosphorylation of 14-3-3 targets.

A developing theme in EC coupling regulation is 14-3-3-dependent strengthening or prolonging of PKA effects, and it is tempting to speculate that this is a conserved mechanism across all cardiac PKA targets. We have recently shown 14-3-3 regulates $Ca_V1.2$ expression and function during $\beta$AR signalling, which suggests an intricate balance between 14-3-3, Rad and $Ca_V1.2$, given that Rad binds to both $Ca_V1.2$ and 14-3-3. On the contrary regulation of the well-known PKA targets RyR and HCN4 have not been investigated to our knowledge and would add to the tapestry of interwoven 14-3-3 regulation of myocyte function. 14-3-3 regulation of gene expression and pathological remodelling presents both an exciting avenue for future therapeutic development and a challenge to make

a therapeutic targeted enough to not cause more harm than good. Isoform specificity and relative abundances, tissue targeting and regulation of 14-3-3 binding partners will need to be carefully considered.

## Supplementary Material

Refer to Web version on PubMed Central for supplementary material.

## Acknowledgements

We are grateful to Mr. Joshua Tulman who produced the artwork featured in the Abstract Figure and Figs 1 and 2.

### Funding

This work was supported by National Institutes of Health grants T32GM099608 and F31HL165815 (H.C.S.) and by R01HL159304 and R01AG063796 (R.E.D.).

## Biography

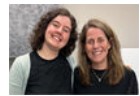

**Dr. Heather C. Spooner** (left) is a postdoctoral scholar in Dr. Karen Zito's lab at UC Davis's Department of Neurobiology, Physiology and Behavior. A California Polytechnic State University graduate, she completed her PhD in Dr. Dixon's lab, studying 14-3-3-mediated regulation of $Ca_V1.2$. She enjoys hiking and skiing in her free time. **Dr. Rose E. Dixon** (right) is an associate professor in UC Davis's Department of Physiology and Membrane Biology. Her research focuses on $Ca_V1.2$ channel trafficking and excitation–contraction coupling in the heart, examining effects of stress, ageing and lipid signalling. After graduating from Queen's University Belfast, she completed her PhD at the University of Nevada and postdoctoral training at the University of Washington. Outside work, she's a passionate sideline supporter at her sons' soccer matches and enjoys snowboarding with her family at nearby Lake Tahoe.

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

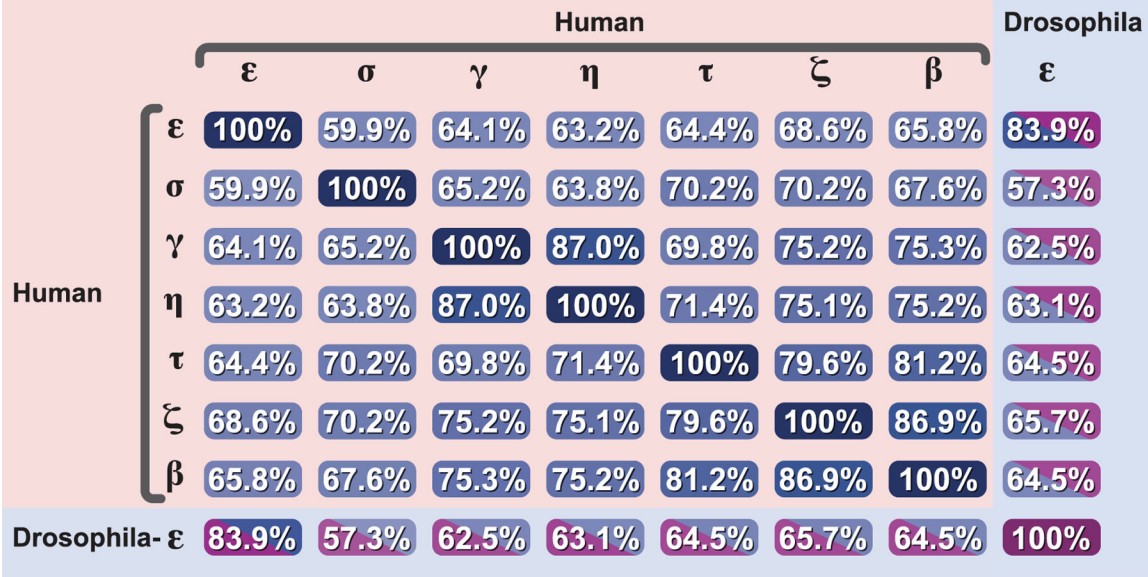

**Figure 1. Conservation analysis of 14-3-3 protein isoforms**

Sequence identity matrix comparing the seven human 14-3-3 isoforms ($\beta$, $\epsilon$, $\gamma$, $\eta$, $\tau$, $\zeta$ and $\sigma$, shown in blue) and *Drosophila melanogaster* 14-3-3$\epsilon$ (shown in green). Values represent percentage amino acid sequence identity between pairs of isoforms, highlighting the high degree of conservation among family members. Notably some human isoforms share greater sequence identity with their corresponding isoforms in other species than with other human isoforms. Analysis performed using UniProt sequence alignment tools.

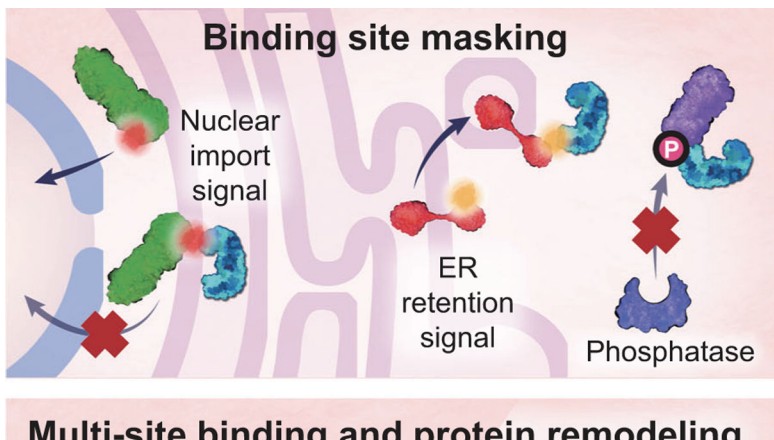

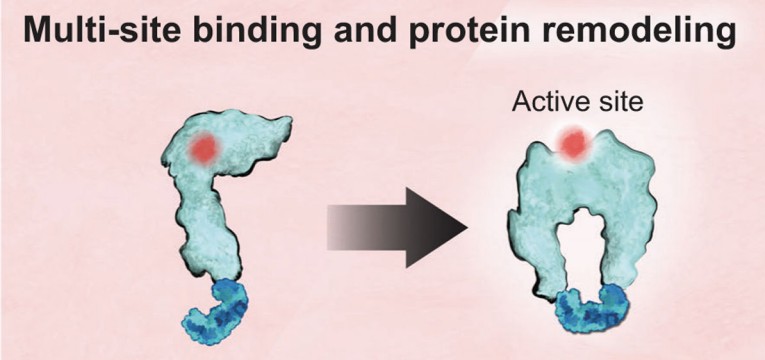

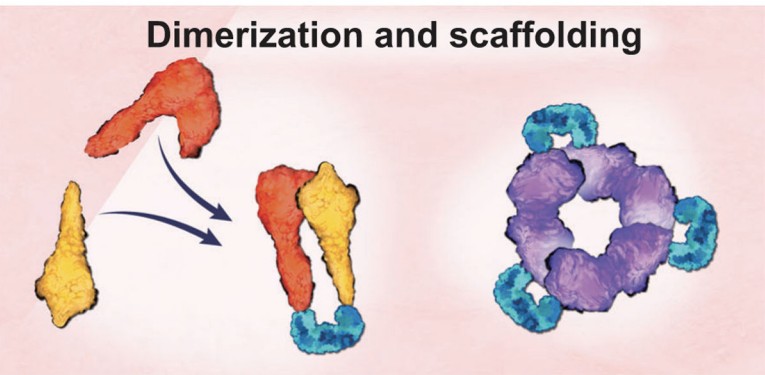

**Figure 2. Core mechanisms of 14-3-3 protein function**

Schematic illustration of the three principal mechanisms by which 14-3-3 proteins regulate their targets: (*top*) masking of binding sites or regulatory motifs to control protein localization, interactions and/or trafficking; (*middle*) clamping through multisite binding to induce or stabilize specific protein conformations or to reveal active sites; and (*bottom*) scaffolding and dimerization to facilitate protein–protein interactions or complex assembly.

