## [Peer Review File · The Journal of physiology]

14-3-3 Proteins: Regulators of Cardiac Excitation- Contraction Coupling and Stress Responses

Heather C. Spooner and Rose E. Dixon
DOI: 10.1113/JP288566

Corresponding author(s): Rose Dixon (redickson@ucdavis.edu)

The following individual(s) involved in review of this submission have agreed to reveal their identity: Fabien Brette (Referee #2)

Review Timeline:

Submission Date:	22-Jan-2025
Editorial Decision:	03-Mar-2025
Revision Received:	31-Mar-2025
Accepted:	14-Apr-2025

Senior Editor: Bjorn Knollmann

Reviewing Editor: Michelle Collins

Transaction Report:

Dear Dr Dixon,

Re: JP-TR-2025-288566 "**14-3-3 Proteins: Regulators of Cardiac Excitation-Contraction Coupling and Stress Responses**" by Rose E. Dixon and Heather C. Spooner

Thank you for submitting your manuscript to The Journal of Physiology. It has been assessed by a Reviewing Editor and by 2 expert referees and we are pleased to tell you that it is acceptable for publication following satisfactory revision.

ABSTRACT FIGURES: Authors may use The Journal's premium BioRender account to create/redraw their Abstract Figures (and any other suitable schematic figure). Information on how to access this account is here: <https://physoc.onlinelibrary.wiley.com/journal/14697793/biorender-access>.

REVISION CHECKLIST: Upload a full Response to Referees file. To create your 'Response to Referees' copy all the reports, including any comments from the Senior and Reviewing Editors, into a Microsoft Word, or similar, file and respond to each point, using font or background colour to distinguish comments and responses and upload as the required file type.

We look forward to receiving your revised submission.

Yours sincerely,

Bjorn Knollmann
Senior Editor

EDITOR COMMENTS

Reviewing Editor:

The manuscript is clearly written and addresses an interesting aspect of cardiac biology. The reviewers identified minor points that should be revised to improve the clarity in some sections.

Please also see 'Required Items' below.

Senior Editor:

I concur with the reviewing editor. Please incorporate the reviewers' suggestions in the revision.

REFEREE COMMENTS

Referee #1:

This a useful review and nicely covers 14-3-3 protein physiology with an emphasis on the myocardium.

I have a few comments and suggestions for the authors.

Line 103: Do isoforms segregate with tissue expression, even if incompletely? In other words, are some tissues enriched in particular isoforms? It would be good for the reader to note isoform/tissue expression here.

Line 375-376: It is noted that 14-3-3 contribute to nuclear sequestration of CaVbeta by RGK proteins; however, there is no evidence that in the myocardium that RGK proteins reside in the nucleus. Thus, it would be good to note the caveat of over-expression / heterologous expression systems and how uncontrolled stoichiometries might flavor data interpretation. Second, it is unclear whether RGK-CaVbeta interactions can dislodge CaVbeta - CaValpha interactions. Early studies noted here are important foundational information - I suggest tempering the conclusion that 14-3-3 might work by freeing CaVbeta subunits to promote channel trafficking.

Line 412, end of line, change 'heart' to 'ventricle.' The specialized conduction system is below the AV node and promotes electrical synchronization for the ventricles.

Line 419 relates studies of 14-3-3 on connexin 43 function. This paragraph should be clarified because the early part the paragraph (starting with line 408) discusses the specialized conduction system of the heart (also called Purkinje fibers). Purkinje fibers express mainly connexin 40 and 45, but not 43. Connexin 43 is important for conduction within the atria and within the ventricle. The only caveat is that distal conducting cardiomyocytes might express connexin 43. The authors should consider refocusing this paragraph removing mention of the conduction system and focusing on the importance of gap junctions for synchronization of myocardial electrical activity.

Line 464-474 raise interesting (& appropriate) conjecture about 14-3-3 regulation via CaV1.2 complex, CaVbeta and Rad. In this vein, the Rad C-terminus contains a polybasic domain long known to be required for RGK-membrane association (Heo et al., 2006 Science 314:1458-1461). In cardiac ventricular cardiomyocytes, loss of the Rad C-terminus, including 14-3-3 interaction sites, causes loss of T-tubule localization and abrogation of LTCC inhibition (Elmore et al, JGP 2024). Thus the

authors should consider the prediction that 14-3-3 regulates t-tubule localization of Rad, in turn providing for 14-3-3 regulation of Rad-LTCC complex.

Line 475: ...'most of the Ca²⁺'; 'most' is certainly true in mice; however, humans and large mammals Ca source can vary substantially. Consider qualifying, or simply starting the paragraph with, 'The SR is a major contributor to contractile calcium.'

Line 481 and this paragraph: This paragraph would benefit from editing. Change ' what goes in must come out on a beat-to-beat' to 'over steady-state transmembrane efflux/ influx as well as trans-SR efflux/influx must balance. Some of this comment is driven by a bit too much use of colloquialisms, but also it is a good place for the authors to remind readers of the finding that for example, upon perturbation of steady state by PKA activation, SR load can change, at least transiently.

Line 486: The authors discuss the interesting documentation of 14-3-3 regulation of PLB-SERCA. Micropeptides, eg, DWORF also regulate SERCA activity. Is there any evidence that 14-3-3 interacts with micropeptides relevant to SERCA activity in cardiomyocytes?

Line 513, Delete, "Perhaps the most famous of the cardiac potassium channels is..." Start sentence with, 'The hERG channel ...

The phrase beginning with 'Perhaps' is distracting.

Referee #2:

This commissioned review summarizes literature on the role of 14-3-3 in cardiac physiology and pathologies. It is very well organized and provides scholarly references. I enjoyed reading it and so will many readers of The Journal.

I have no major comments.

I just noticed one missing point

The SWTY motif which is functionally interchangeable with a known motif in cardiac potassium channels (Kir2.1) and operates by recruiting 14-3-3 proteins. (Nat Cell Biol 2005 PMID: 16155591). In a more recent study, it was shown that inhibition of 14-3-3 proteins did not modify the INa and IK1 densities generated by each channel separately, whereas it decreased the INa and IK1 generated when they were co-expressed (Front Physiol 2017 PMID: 29184507). I think it worth adding and discussing these publications.

REQUIRED ITEMS

- Please include an Abstract Figure file, as well as the Figure Legend text within the main article file. The Abstract Figure is a piece of artwork designed to give readers an immediate understanding of the Review Article and should summarise the main conclusions. If possible, the image should be easily 'readable' from left to right or top to bottom. It should show the physiological relevance of the Review so readers can assess the importance and content of the article. Abstract Figures should not merely recapitulate other figures in the Review. Please try to keep the diagram as simple as possible and without superfluous information that may distract from the main conclusion of the Review. Abstract Figures must be provided by authors no later than the revised manuscript stage and should be uploaded as a separate file during online submission

labelled as File Type 'Abstract Figure'. Please ensure that you include the figure legend in the main article file. All Abstract Figures will be sent to a professional illustrator for redrawing and you may be asked to approve the redrawn figure before your paper is accepted.

- Please upload separate high quality figure files via the submission form.

- Author profile(s) must be uploaded via the submission form. Authors should submit a short biography (no more than 100 words for one author or 150 words in total for two authors) and a portrait photograph of the two leading authors on the paper. These should be uploaded and clearly labelled together in a Word document with the revised version of the manuscript. Any standard image format for the photograph is acceptable, but the resolution should be at least 300 DPI and preferably more. A group photograph of all authors is also acceptable, providing the biography for the whole group does not exceed 150 words.

- Please ensure that the Article File you upload is a Word file.

END OF COMMENTS

Responses to the Reviewers Comments

We thank the reviewers for their time and effort in reviewing our review on 14-3-3 Proteins: Regulators of Cardiac Excitation-Contraction Coupling and Stress Responses. We are encouraged by the positive reception regarding the clarity of our writing and the relevance of our topic to cardiac biology. The reviewers' feedback has been invaluable in refining our work. We have thoroughly addressed each suggestion and have implemented appropriate changes throughout the manuscript. Detailed responses to each reviewer comment are provided below.

Reviewer 1's comments:

1) This a useful review and nicely covers 14-3-3 protein physiology with an emphasis on the myocardium.

We thank the referee for these kind comments.

2) Line 103: Do isoforms segregate with tissue expression, even if incompletely? In other words, are some tissues enriched in particular isoforms? It would be good for the reader to note isoform/tissue expression here.

Thank you for this insightful suggestion. While tissue-specific 14-3-3 isoform expression patterns are complex (as we discuss later in the review), we agree that providing this context earlier would better orient readers. To address this point, we have made the following additions:

- At line 74, we added:

"In humans, seven distinct isoforms (β , ϵ , γ , η , τ , ζ , and σ) are encoded by genes located on different chromosomes, all of which are expressed in the healthy heart with the exception of 14-3-3 σ ."

- In the section on 14-3-3 expression, we expanded with a new paragraph:

"While a recent review summarized the current knowledge of human 14-3-3 protein isoform distribution across multiple tissues, the heart was notably excluded from this analysis {Sengupta, 2020 #22}. Here, we highlight the present understanding of cardiac 14-3-3 isoform expression, which exhibits some species-specific variations. In rodents, RT-PCR analysis of rat cardiac tissue revealed an mRNA abundance hierarchy of YWHAE>YWHAH>YWHAZ>YWHAB >YWHAG>YWHAQ that held when comparing atrial with

ventricular samples {Arakel, 2014 #46}. This pattern largely mirrors the hierarchy of mouse heart protein isoform expression ($\epsilon > \gamma > \zeta > \beta > \eta > \tau$), particularly at the extremes of abundance as quantified through SILAC proteome analysis {Qu, 2022 #253}. Human expression profiles, however, show inconsistencies across studies. Analysis of human transcriptome data (BioProject: PRJNA264807; GSE62689) by Qu and colleagues identified an abundance pattern of YWHAG>YWHAE>YWHAB> YWHAH>YWHAZ>YWHAQ, while Thompson and Goldspink, using the Human Protein Atlas consensus dataset (www.proteinatlas.org), reported a different expression hierarchy: YWHAQ> YWHAE>YWHAB>YWHAG>YWHAZ>YWHAH {Thompson, 2022 #37}. The variations observed in human samples may reflect not only methodological differences but also genuine individual-to-individual variations in isoform expression, potentially influenced by factors such as age, sex, genetic background, and underlying health conditions. A systematic analysis accounting for these variables could help establish more definitive expression patterns and potentially identify clinically relevant expression signatures."

These additions highlight that while comprehensive reviews of 14-3-3 isoform distribution exist for various tissues, the heart has been notably overlooked in previous analyses. Our expanded discussion provides the tissue-specific context requested by the reviewer while maintaining the review's logical flow and strengthening its focus on cardiac biology specifically.

3) Line 375-376: It is noted that 14-3-3 contribute to nuclear sequestration of CaVbeta by RGK proteins; however, there is no evidence that in the myocardium that RGK proteins reside in the nucleus. Thus, it would be good to note the caveat of over-expression / heterologous expression systems and how uncontrolled stoichiometries might flavor data interpretation. Second, it is unclear whether RGK-CaVbeta interactions can dislodge CaVbeta - CaValpha interactions. Early studies noted here are important foundational information - I suggest tempering the conclusion that 14-3-3 might work by freeing CaVbeta subunits to promote channel trafficking.

This point is well taken. In response we rewrote and expanded this section of the review to incorporate this correct information. The section now reads:

"An illustrative case involves the reported interactions between 14-3-3, RGK proteins (Rad, Rem, Rem2, and Gem/Kir), and Ca²⁺ channel auxiliary subunits. Evidence from heterologous

expression studies, suggests that RGK proteins such as Rad and Rem can sequester calcium channel β subunits, and that 14-3-3 binding to RGK proteins might interfere with this interaction (Beguin *et al.*, 2006). However, it is important to note several caveats: (1) to our knowledge, there is no evidence demonstrating nuclear localization of RGK proteins in native cardiomyocytes under physiological conditions; (2) heterologous expression systems with non-physiological protein stoichiometries may not accurately reflect *in vivo* interactions; and (3) it remains unclear whether RGK- $\text{Ca}_v\beta$ interactions are sufficiently strong to disrupt established $\text{Ca}_v\beta$ - $\text{Ca}_v\alpha$ interactions in the physiological context. While these studies provide important foundational insights into potential regulatory mechanisms, further investigation in cardiac-specific contexts with appropriate stoichiometries is needed to establish whether 14-3-3 proteins indeed modulate Ca^{2+} channel trafficking through this pathway in the heart."

4) Line 412, end of line, change 'heart' to 'ventricle.' The specialized conduction system is below the AV node and promotes electrical synchronization for the ventricles.

Changed.

5) Line 419 relates studies of 14-3-3 on connexin 43 function. This paragraph should be clarified because the early part of the paragraph (starting with line 408) discusses the specialized conduction system of the heart (also called Purkinje fibers). Purkinje fibers express mainly connexin 40 and 45, but not 43. Connexin 43 is important for conduction within the atria and within the ventricle. The only caveat is that distal conducting cardiomyocytes might express connexin 43. The authors should consider refocusing this paragraph removing mention of the conduction system and focusing on the importance of gap junctions for synchronization of myocardial electrical activity.

The early part of this paragraph was meant as an introduction to EC coupling; the implication of connexin 43 being highly expressed in the conduction system itself was unintended and we appreciate the referee bringing this to our attention. We have edited this section for clarity and now start the paragraph with:

"A crucial element of cell-to-cell conduction in myocytes is the gap junctions, which are low resistance channels that allow cell-to-cell coupling of adjacent cardiomyocytes at their borders, called intercalated disks".

6) Line 464-474 raise interesting (& appropriate) conjecture about 14-3-3 regulation via CaV1.2 complex, CaVbeta and Rad. In this vein, the Rad C-terminus contains a polybasic domain long known to be required for RGK-membrane association (Heo et al., 2006 Science 314:1458-1461). In cardiac ventricular cardiomyocytes, loss of the Rad C-terminus, including 14-3-3 interaction sites, causes loss of T-tubule localization and abrogation of LTCC inhibition (Elmore et al, JGP 2024). Thus, the authors should consider the prediction that 14-3-3 regulates t-tubule localization of Rad, in turn providing for 14-3-3 regulation of Rad-LTCC complex.

We thank the referee for raising this important point. We had not yet considered this possibility and have now expanded this section to include a larger discussion of this topic. Adding the following:

" The Satin group recently developed an innovative mouse model featuring truncation of Rad's polybasic C-terminal region (Elmore *et al.*, 2024). This positively charged polybasic domain, found in Rad and other RGK proteins, normally facilitates membrane localization through electrostatic interactions with negatively charged phosphatidylinositol lipids (Heo *et al.*, 2006{Correll, 2007 #577}). The truncation causes Rad to mislocalize to the cytosol instead of its normal t-tubule position, disrupting its interaction with Ca_vα_{1C}-associated Ca_vβ and precluding both its basal suppression of *I*_{Ca} and its β-adrenergic signaling-triggered augmentation (Elmore *et al.*, 2024). Notably, this truncation removes the S300 14-3-3 binding site on Rad (Finlin & Andres, 1999; Beguin *et al.*, 2006; Spooner *et al.*, 2025). Previous studies have linked 14-3-3 expression to RGK protein redistribution from the nucleus to the cytosol in a mechanism requiring the intact Rad C-terminal (Beguin *et al.*, 2006). These observations, when viewed alongside the recent Satin lab findings, raise the intriguing, though as yet unexplored, possibility that 14-3-3 might contribute to Rad localization in cardiomyocytes. This hypothesis that warrants further investigation.

Marx and colleagues further demonstrated that Rad must dissociate from both the membrane (Papa *et al.*, 2024) and from Ca_vβ to permit β-adrenergic regulation of Ca_v1.2 channels (Liu *et al.*, 2020). This process depends on PKA-mediated phosphorylation of two specific C-terminal serine residues (S272 and S300), which generates electrostatic repulsion from the negatively charged membrane (Yang *et al.*, 2019; Papa *et al.*, 2024). Charge-substitution experiments with aspartic acid residues confirmed this mechanism, showing relief of Rad's suppressive effect on channel open probability with increasing negative charge (Papa *et*

al., 2024). Notably, membrane-anchored Rad (via CAAX fusion) remains locked in its inhibitory state despite phosphorylation or charge substitutions (Papa *et al.*, 2024).

The potential role of 14-3-3 proteins in this system is particularly intriguing since Ser300 is a known 14-3-3 binding site. While 14-3-3 preferentially binds to phosphorylated serines and threonines, it can also interact with aspartic acid residues (Petosa *et al.*, 1998), suggesting 14-3-3 might contribute to the effects observed with charge-substituted Rad mutants. Competition studies with the related RGK protein Kir/Gem revealed mutually exclusive binding relationships with 14-3-3, calmodulin (CaM), and $Ca_v\beta$, ie. Kir/Gem can only bind one of those interaction partners at a time (Beguin *et al.*, 2005). If Rad exhibits similar exclusivity, 14-3-3 binding to phosphorylated or charge-substituted Rad could sequester it from its low-affinity interaction with $Ca_v\beta$ (Xu *et al.*, 2015), contributing to channel disinhibition. Membrane anchoring might sterically block 14-3-3 access to these binding sites, explaining the locked inhibitory state of CAAX-Rad fusion proteins. Understanding these complex interaction networks will be crucial for uncovering the full extent of Rad-dependent and Rad-independent regulation of $Ca_v1.2$ by 14-3-3."

7) Line 475: ...'most of the Ca^{2+} '; 'most' is certainly true in mice; however, humans and large mammals Ca source can vary substantially. Consider qualifying, or simply starting the paragraph with, 'The SR is a major contributor to contractile calcium.'

Fair point. In response we revised this sentence to: "On the other side of the dyad, ryanodine receptors (RyR2) serve as crucial SR calcium release channels."

8) Line 481 and this paragraph: This paragraph would benefit from editing. Change ' what goes in must come out on a beat-to-beat' to 'over steady-state transmembrane efflux/ influx as well as trans-SR efflux/influx must balance. Some of this comment is driven by a bit too much use of colloquialisms, but also it is a good place for the authors to remind readers of the finding that for example, upon perturbation of steady state by PKA activation, SR load can change, at least transiently.

The language of this section has been clarified and formalized, and now reads:

" Ca^{2+} homeostasis is essential for proper cardiomyocyte function. Over the steady state, Ca^{2+} influx and efflux must balance as must SR Ca^{2+} release and reuptake (Eisner *et al.*, 2013).

Without this equilibrium, myocytes would either accumulate Ca^{2+} and be unable to relax, or gradually deplete their Ca^{2+} stores and lose contractile force. Of course, when regulatory pathways such as β -adrenergic signaling alter Ca^{2+} handling, transient imbalances can occur, leading to changes in SR calcium load before a new steady state is established. SR calcium reuptake is performed by SERCA, a calcium pump that is inhibited by PLB. When PLB is phosphorylated by PKA, it detaches from SERCA, relieving the inhibition. 14-3-3 prolongs the phosphorylated state of PLB (Menzel *et al.*, 2020), shielding it from dephosphorylation. This protection mechanism prolongs SERCA's enhanced activity, resulting in both accelerated relaxation and augmented contractility through increased SR calcium stores and subsequent heightened calcium release during excitation-contraction coupling."

9) Line 486: The authors discuss the interesting documentation of 14-3-3 regulation of PLB-SERCA. Micropeptides, eg, DWORF also regulate SERCA activity. Is there any evidence that 14-3-3 interacts with micropeptides relevant to SERCA activity in cardiomyocytes?

The authors are unaware of any evidence related to 14-3-3 interacting with DWORF micropeptides, although we agree this would be an interesting direction to investigate.

Line 513, Delete, "Perhaps the most famous of the cardiac potassium channels is..." Start sentence with, 'The hERG channel ...

The phrase beginning with 'Perhaps' is distracting.

Agreed and changed as suggested.

Reviewer 2's comments:

This commissioned review summarizes literature on the role of 14-3-3 in cardiac physiology and pathologies. It is very well organized and provides scholarly references. I enjoyed reading it and so will many readers of The Journal.

We thank the referee for these kind comments.

I have no major comments. I just noticed one missing point. The SWTY motif which is functionally interchangeable with a known motif in cardiac potassium channels (Kir2.1) and operates by recruiting 14-3-3 proteins. (Nat Cell Biol 2005 PMID: 16155591). In a more recent study, it was shown that Inhibition of 14-3-3 proteins did not modify the INa and IK1 densities generated by each channel separately, whereas it decreased the INa and IK1 generated when they were co-expressed (Front Physiol 2017 PMID: 29184507). I think it worth adding and discussing these publications.

These are indeed very relevant papers, and we thank the referee for bringing them to our attention. To address this critique, we have added the following paragraphs to the relevant sections.

Accordingly, in the section on trafficking we now state:

"Beyond COPI-mediated retention, 14-3-3 proteins also facilitate forward trafficking by overcoming other ER retention signals. A notable example is the RKR motif, an arginine-based ER retention signal found in various membrane proteins. A screen performed to identify signals capable of overcoming this ER-retention sequence identified the SWTY motif as restoring surface expression to Kir2.1 channels fused to RKR (Shikano *et al.*, 2005). This forward trafficking mechanism requires phosphorylation-dependent recruitment of 14-3-3, which effectively masks the ER-retention signal. SWTY-like motifs have been discovered in several proteins, including the cardiac TASK-1 K⁺ channel (Shikano *et al.*, 2005), suggesting this may be a widespread regulatory mechanism. For other cardiac membrane proteins like the L-type Ca²⁺ channel Ca_v1.2, 14-3-3 has been shown to influence trafficking, though the precise mechanism remains under investigation, with ER retention signal masking representing a plausible mechanism (Spooner *et al.*, 2025)".

In the section on sodium channels, we now state:

"Expanding our understanding of 14-3-3's regulation of cardiac sodium channels, another 2017 study demonstrated that a dominant negative 14-3-3 η mutant (R56, 60A) selectively affected Na_v1.5 currents in CHO cells only when co-expressed with Kir2.1, but had no effect when either channel was expressed individually (Utrilla *et al.*, 2017). This finding suggests that these adaptor proteins may orchestrate the coordinated activity of different ion channel types that work in concert to shape the cardiac action potential."

They have been added to the trafficking, sodium channel, and potassium channel sections.

REQUIRED ITEMS

- Please include an Abstract Figure file, as well as the Figure Legend text within the main article file. The Abstract Figure is a piece of artwork designed to give readers an immediate understanding of the Review Article and should summarise the main conclusions. If possible, the image should be easily 'readable' from left to right or top to bottom. It should show the physiological relevance of the Review so readers can assess the importance and content of the article. Abstract Figures should not merely recapitulate other figures in the Review. Please try to keep the diagram as simple as possible and without superfluous information that may distract from the main conclusion of the Review. Abstract Figures must be provided by authors no later than the revised manuscript stage and should be uploaded as a separate file during online submission labelled as File Type 'Abstract Figure'. Please ensure that you include the figure legend in the main article file. All Abstract Figures will be sent to a professional illustrator for redrawing and you may be asked to approve the redrawn figure before your paper is accepted.

Done.

- Please upload separate high quality figure files via the submission form.

Done.

- Author profile(s) must be uploaded via the submission form. Authors should submit a short biography (no more than 100 words for one author or 150 words in total for two authors) and a portrait photograph of the two leading authors on the paper. These should be uploaded and clearly labelled together in a Word document with the revised version of the manuscript. Any standard image format for the photograph is acceptable, but the resolution should be at least

300 DPI and preferably more. A group photograph of all authors is also acceptable, providing the biography for the whole group does not exceed 150 words.

Done.

- Please ensure that the Article File you upload is a Word file.

Done.

Dear Dr Dixon,

Re: JP-TR-2025-288566R1 "**14-3-3 Proteins: Regulators of Cardiac Excitation-Contraction Coupling and Stress Responses**" by Heather C. Spooner and Rose E. Dixon

We are pleased to tell you that your paper has been accepted for publication in The Journal of Physiology.

Authors should note that it is too late at this point to offer corrections prior to proofing. Major corrections at proof stage, such as changes to figures, will be referred to the Editors for approval before they can be incorporated. Only minor changes, such as to style and consistency, should be made at proof stage. Changes that need to be made after proof stage will usually require a formal correction notice.

Yours sincerely,

Bjorn Knollmann
Senior Editor
The Journal of Physiology

P.S. - You can help your research get the attention it deserves! Check out Wiley's free Promotion Guide for best-practice recommendations for promoting your work at www.wileyauthors.com/eoo/guide. You can learn more about Wiley Editing Services which offers professional video, design, and writing services to create shareable video abstracts, infographics, conference posters, lay summaries, and research news stories for your research at www.wileyauthors.com/eoo/promotion.

IMPORTANT NOTICE ABOUT OPEN ACCESS: To assist authors whose funding agencies mandate public access to published research findings sooner than 12 months after publication, The Journal of Physiology allows authors to pay an Open Access (OA) fee to have their papers made freely available immediately on publication.

You can check if your funder or institution has a Wiley Open Access Account here: <https://authorservices.wiley.com/author-resources/Journal-Authors/licensing-and-open-access/open-access/author-compliance-tool.html>.

EDITOR COMMENTS

Reviewing Editor:

All comments were sufficiently addressed.

Senior Editor:

The manuscript is now acceptable. Thank you for your excellent contribution to the Journal!